# Carbon Dioxide Emissions Reduction through Technological Innovation: Empirical Evidence from Chinese Provinces

**DOI:** 10.3390/ijerph19159543

**Published:** 2022-08-03

**Authors:** Yanran Liu, Lei Tang, Guangfu Liu

**Affiliations:** 1Shanghai International College of Intellectual Property, Tongji University, Shanghai 200092, China; yanranliu@tongji.edu.cn; 2School of Economics and Management, Tongji University, Shanghai 200092, China; gfuliu@tongji.edu.cn

**Keywords:** carbon emission reduction, technological innovation, spatial econometrics, spatial mediation model, empirical analysis

## Abstract

Energy consumption and industrial activities are the primary sources of carbon emissions. As the “world’s factory” and the largest carbon emitter, China has been emphasizing the core role of technological innovation in promoting industrial structure upgrades (ISU) and energy efficiency (EE) to reduce carbon emissions from industrial production and energy consumption. This study investigated the mechanism (through ISU and EE) and spillover effect of technological innovation on carbon emission reduction using the panel dataset of 30 Chinese provinces from 2008 to 2019 and spatial econometrics models. The study concluded that (1) technological innovation had a negative direct effect on provincial carbon emissions, while it also showed a spatial spillover effect on neighboring provinces; (2) technological innovation had an indirect effect on provincial carbon emissions reduction through the mediation of energy efficiency improvement, while the mediation effect of industrial structure upgrading is not yet significant; and (3) the effect of technological innovation on carbon emission reduction showed heterogeneity in the eastern, central, and western regions of China. This study provided empirical and theoretical references to decision-makers in China and other developing countries in promoting technological and carbon control policies. More specifically, direct technology investment and indirect investment in industrial structure upgrades and energy efficiency could help with regional carbon emissions reduction.

## 1. Introduction

Since the industrial revolution, global climate change has been characterized by warming. The increasing demand for natural resources from human production and living activities has led to massive carbon dioxide (CO_2_) greenhouse gas emissions and ultimately negative impacts on the Earth’s ecology [1,2]. Due to global warming, extreme weather events such as glacial melting, forest fires, intense hurricanes, floods, extreme cold in winter, and extreme heatwaves in summer have occurred frequently in recent years [3,4]. The frequency of extreme events underscores the urgency of taking action to achieve net-zero CO_2_ emissions to improve environmental quality. By the middle of the 21st century, achieving net-zero CO_2_ emissions and promoting sustainable human development have become common goals the international community pursues.

China plays a leading role in global green development as the largest developing economy. The Chinese government has been committed to carbon emission reduction and has incorporated the “dual carbon” goal into the national 14th Five-Year Plan and the 2035 Vision. The Intergovernmental Panel on Climate Change (IPCC) assessment report states that innovation and technological advances are critical to reducing carbon emissions. The scale and speed of technological development will determine the pace of future carbon reductions [5]. However, previous literature on the relationship between technological innovation and carbon emissions showed mixed conclusions due to different perspectives and research methodologies. Therefore, it is essential to explore further whether technological innovation promotes or hinders carbon emissions, whether these effects have a spatial spillover effect on neighboring regions, and the mechanism through which technological progress could assist with regional carbon emissions reduction. Moreover, it is also crucial to identify the heterogeneity in the effects due to the significant differences in regional resource endowments. In this context, this study used spatial econometrics models to empirically analyze the intrinsic mechanism of technological innovation on carbon emissions from a provincial and a regional perspective and explored the effect of technological innovation on regional carbon emission reduction.

The structure of the study is shown in Figure 1. Firstly, we used a DEA-Malmquist method to calculate the total factor productivity of 30 Chinese provinces as an indicator of technological innovation, where R&D investment and labor force were used as input variables, and GDP was used as a desired output variable. Secondly, the spatial econometric models were applied to analyze the relationship between carbon emissions and technological innovation. In addition, the spatial mediation model was applied to investigate how technological innovation affects carbon emissions using industrial structure upgrades and energy efficiency improvements as mediators. Finally, we test the regional heterogeneity in the eastern, central, and western regions.

## 2. Literature Review

The study of the key impact factors of carbon emissions is one of the major research hotspots at present. Scholars pointed out that economic development [6], urbanization [7], foreign direct investment [8], energy consumption [9], energy structure [10], and industrial structure [11] are all potential drivers of carbon emissions. With the development of scientific research and social progress, technological innovation has been recognized as one of the critical drivers of economic growth [12] and the core of climate control [13]. Li Shasha et al. [14]) used a static panel model and a dynamic panel model to examine the effect of technological progress on carbon emissions and found that technological progress can significantly suppress CO_2_ emissions.

It is important to note that technological innovation and carbon emissions are closely related in neighboring regions of China regarding resource sharing, technology exchange, economic development, and policies. Therefore, the environmental improvements brought about by technological innovation in one region are bound to impact the carbon emissions of neighboring provinces. When studying the impact of technological innovation on interregional carbon emissions, the results may be biased if the spatial spillover effect is ignored [15]. With the expanding application of spatial econometrics, many scholars have considered including spatial effects in studies related to technological innovation and carbon emissions [16,17,18].

Scholars have explored the impact of technological innovation on carbon emissions and have not yet reached a consensus. By exploring the relationship between the effects of endogenous technologies on carbon emissions, Jaffe et al. [19] concluded that technological innovation may either increase or decrease carbon emissions. Most scholars believe that technological innovation contributes to the reduction of CO_2_ emissions and improvement of environmental quality [20,21] and explain accordingly. For example, strengthening environmental regulations has contributed to the increasing number of direct environmental innovations for carbon emission reduction [22], which have effectively advanced the application of new technologies, thus directly improving energy efficiency and reducing energy consumption. In addition, technological innovation contributes significantly to economic restructuring and optimization by transforming traditional economic development that relies on production factors into an innovation-driven model, which reduces the CO_2_ emissions caused by industrialization [23]. However, scholars who hold the opposite view argue that technological progress can improve resource use efficiency with a diminishing marginal effect, thus creating more demands on natural resources and energy. Due to the rapidly growing economic size, technological advances will either show an increasing or insignificant impact on carbon emissions [24,25].

In summary, the view that technological innovation affects carbon emissions is controversial and deserves further exploration. First, when analyzing panel samples, relevant studies rarely consider regional carbon emissions’ spatial dependence or spillover effects. Second, the intrinsic mechanism through which technological innovation affects carbon emissions lacks empirical analysis. Therefore, this study uses spatial econometric models to empirically investigate technological innovation’s direct and indirect effect on promoting carbon emission reduction in Chinese provinces.

The contribution of this study is mainly reflected in three aspects: First, through the investigation of the direct and indirect effects, we were able to shed light on the debate on the relationship between technological innovation and carbon emissions at the provincial and regional levels. Second, through the spatial mediation model, we were able to explore the intermediary transmission mechanisms of industrial structure upgrading and energy efficiency improvements in promoting technology-driven carbon emission reduction. Third, considering heterogeneities in resource endowment, technology level, and economic level among Chinese provinces, we could identify the differences in the effects of technological innovation on carbon emission reduction in the eastern, central, and western regions. This research provides a theoretical and practical reference to regional decision-makers of technology and carbon control policies in different development stages.

## 3. Hypotheses Development

From the geographic perspective of spatial interaction in the Chinese provinces, technological innovation drives regional carbon emission governance by accumulating and transferring reticent knowledge or sticky knowledge in carbon emission reduction and control. Developing and applying green and low-carbon technology also promote carbon emission reduction. Provinces continuously learn from and imitate neighboring regions to collect resources and knowledge for technological innovation. Meanwhile, areas close to each other in geographical locations are more consistent in regional development levels and technical capabilities; thus, technology innovation in one province could indirectly influence carbon emissions in neighboring regions through the transfer and spillover of knowledge and innovation. Therefore, we proposed that:

**Hypothesis** **1a.***Technological innovation is negatively associated with provincial carbon emissions*.

**Hypothesis** **1b.**
*Technological innovation is negatively associated with carbon emissions in neighboring provinces (a spatial spillover effect).*


Carbon intensity varies by industry. Provinces with different industrial structures may significantly impact regional carbon emissions. The industrial structure determines the allocation of production factors (such as labor, technology, energy, labor, etc.), which in turn determines the emissions and pollution associated with industrial activities. At present, China’s industrial sector is dominated by energy-intensive industries. The upgrading of industrial structure will shift production from low-value-added, high-emission industries to high-value-added, low-carbon industries [26], thereby reducing the proportion of pollution-intensive industries’ output value in the national economy and improving carbon emission efficiency. Technological innovation is the strategic engine for upgrading the industrial structure, improving the technical efficiency of factors, and promoting industrial transformation and upgrading. By studying the Beijing-Tianjin-Hebei urban agglomeration, Runde Gu et al. [27] found that government expenditure on science and technology can promote the upgrading of industrial structure to a certain extent, thereby reducing carbon emissions. Therefore, we propose that:

**Hypothesis** **2.***The negative impact of technological innovation on carbon emissions is mediated by industrial structure upgrades*.

Energy consumption is the primary source of carbon emissions. China’s energy carbon emissions account for more than 90%, and industrial carbon emissions account for more than 70% of energy consumption carbon emissions. Energy efficiency refers to the ratio of the amount of energy utilized to the amount of energy actually consumed. Improving energy efficiency is generally through adopting more efficient technologies or production processes or applying generally accepted methods of reducing energy losses. Technological innovation promotes energy efficiency improvement, which reduces the energy required to provide products or services of the same level. Through technological innovation, the electric motor system could have a more efficient configuration, thus reducing the energy consumption of the electric motors; at the same time, optimizing the control of the heating and cooling systems through technical approaches could reduce the energy use of the building. Furthermore, through technological innovation, energy can be recovered and reused to improve the overall efficiency of energy consumption, thereby achieving the effect of carbon emission reduction.

Technological innovations in energy conservation and emission reduction can improve energy use efficiency. Green technology innovation in energy enhances the rate of clean energy substitution in production. Many studies have concluded that technological innovation can optimize energy structure and improve carbon emission reduction performance [28]. Therefore, we propose:

**Hypothesis** **3.***The negative impact of technological innovation on carbon emissions is mediated by energy efficiency improvements*.

The regions’ resource endowments, technology capabilities, and economic levels cause significant differences in industrial structure and energy efficiency [29]. From the scale of investment in science and technology, the eastern region is significantly ahead of other regions. The scientific and technological investment intensity (ratio of R&D funds to GDP) in the eastern region is higher than that in the central and western regions; the proportion of R&D achievements and scientific researchers in the eastern region is higher than that in the other two regions, and the number of high-tech industries and the number of patent applications is the highest in the eastern region, and the central region also shows significant growth potential, but that in the western region show a slow growth rate. Therefore, the effect of technological innovation on carbon emission reduction may show heterogeneity. Thus, we propose:

**Hypothesis** **4.***The direct and indirect effects of technological innovation on carbon emissions are different in the eastern, central, and western regions of China*.

## 4. Methodology and Data

### 4.1. Spatial Econometric Model

Technological innovation is non-competitive and partially non-exclusive, and therefore prone to technological spillovers. Innovative activities in one region may benefit other regions not involved in technological innovation. While promoting regional carbon emission management, technological innovation causes neighboring provinces and regions to continuously learn and imitate, indirectly reducing carbon pollution in neighboring provinces and regions. Therefore, ignoring the spatial correlation of technological innovation will lead to inaccurate results. Since different spatial econometric models assume different spatial transmission mechanisms and economic significance, the selection of spatial econometric models is important [30]. In order to obtain a better fit, a spatial econometric model test was conducted by referring to Elhorst [31] before performing the regression, and then the spatial econometric model suitable for this study was selected. The LM and robust LM tests were used first to determine the corresponding fitness of the SEM, SAR, and SDM models. The LR and Wald tests were conducted to determine whether the SDM model would degenerate into a SAR or SEM model. The Hausman test was used to determine whether the random or fixed-effects model was chosen. The testing process followed the path of ordinary least squares (OLS), spatial autoregressive model (SAR), spatial error model (SEM), and spatial Durbin model (SDM). To eliminate the effect of heteroskedasticity, we used the logarithm term of the non-ratio variables.

The OLS model does not consider the existence of spatial dependence between regions, which can easily lead to biased estimation results. The expression is shown in Equation (1).
(1)$$\begin{array}{ll} {\ln CO_{2}}_{it}= & \beta_{0}+\beta_{1}G_{it}+\beta_{2}U_{it}+\beta_{3}\ln FDI_{it}+\beta_{4}\ln PGDP_{it}+\beta_{5}lnEN_{it}+\beta_{6}MD_{it}+\\ & \beta_{7}IN_{it}+\beta_{8}\ln POP_{it}+\varepsilon_{it} \end{array}$$

In the SAR model, the spatial dependence between variables leads to spatial correlation, with a unidirectional spatial correlation between regions. The model contains lagged terms of the spatial dependent variables, as shown in Equation (2).
(2)$$\begin{array}{cc} {\ln{CO}_{2}}_{it}= & \beta_{0}+\delta W\ln{CO_{2}}_{it}+\beta_{1}G_{it}+\beta_{2}U_{it}+\beta_{3}\ln FDI_{it}+\beta_{4}\ln PGDP_{it}+\beta_{5}lnEN_{it}+\\ & \beta_{6}MD_{it}+\beta_{7}IN_{it}+\beta_{8}\ln POP_{it}+\varepsilon_{it} \end{array}$$

In the SEM model, the cause of technological innovation spillover is a random shock. Compared to the OLS model, its spatial effect is mainly transmitted through the error term. The model contains the random error autocorrelation term as the spatial error term. The specific expression is shown in Equation (3).
(3)$$\begin{array}{ll} {\ln CO_{2}}_{it}= & \beta_{0}+\beta_{1}G_{it}+\beta_{2}U_{it}+\beta_{3}\ln FDI_{it}+\beta_{4}\ln PGDP_{it}+\beta_{5}lnEN_{it}+\beta_{6}MD_{it}+\\ & \beta_{7}IN_{it}+\;\beta_{8}\ln POP_{it}+\mu_{it} \end{array}$$
$$\mu_{it}=\lambda W\mu_{it}$$

Regional carbon emissions are influenced not only by local independent variables but also by other regional independent variables. In the SDM model, both the changes in error terms caused by the spatial lag of the dependent variable and the spatial interactions between regions are considered. The specific expression is shown in Equation (4).
(4)$$\begin{array}{ll} \ln{CO_{2}}_{it}= & \beta_{0}+\delta W\ln{CO_{2}}_{it}+\beta_{1}G_{it}+\beta_{2}U_{it}+\beta_{3}\ln FDI_{it}+\beta_{4}\ln PGDP_{it}+\beta_{5}\ln EN_{it}+\\ & \beta_{6}MD_{it}+\beta_{7}IN_{it}+\theta_{1}WG_{it}+\theta_{2}WU_{it}+\theta_{3}W\ln FDI_{it}+\theta_{4}W\ln PGDP_{it}+\\ & \;\theta_{5}W\ln EN_{it}+\theta_{6}WMD_{it}+\theta_{7}WIN_{it}+\theta_{8}W\ln POP_{it}+\varepsilon_{it} \end{array}$$
where ${CO_{2}}_{it}$ represents the provincial carbon emissions; $G_{it}$ is the core explanatory variable, which represents province-level technological innovation. In addition, seven control variables, urbanization level ($U_{it}$), foreign direct investment ($FDI_{it}$), economic development level ($PGDP_{it}$), energy consumption ($EN_{it}$), marketization level ($MD_{it}$), industrialization level ($IN_{it}$), and regional population size ($POP_{it}$), are added to consider provincial characteristics. The definition and sources of each variable are described in detail below. The $\mu_{it}$ term and $\varepsilon_{it}$ term are perturbations that follow an independent uniform distribution. W term is the spatial weight matrix, and this study used the spatial distance weight matrix, which means all main diagonal elements are 0 and all non-main diagonal elements are $1/d^{2}$, where d is the distance between the geographic center locations of two provinces.

### 4.2. Spatial Mediation Model

Technological innovation may affect provincial carbon emissions through industrial structure upgrading and energy efficiency improvement. Therefore, a more normative mediation model must be constructed to confirm Hypotheses 2 and 3. In the spatial mediation models, Y refers to the amount of provincial carbon emissions, M refers to industrial structure upgrades and energy efficiency, and X refers to technological innovation. The relevant control variables are kept consistent with the original model, and then the specific mediating effect test models are set in Equations (4)–(6). The test procedure draws on the stepwise mediating effect test proposed by Baron and Kenny [32], and the steps are shown in Figure 2.
(5)$$ISS_{it}=\alpha_{0}+\alpha_{1}\ln G_{it}+\alpha_{2}X_{control}+\delta_{1}W\ln G_{it}+\delta_{2}WX_{control}+\varepsilon_{it}$$
(6)$$\ln CO_{2it}=\rho_{0}+\rho_{1}\ln G_{it}+\rho_{2}ISS_{it}+\sigma_{1}W\ln G_{it}+\sigma_{2}WISS_{it}+\sigma_{3}WX_{control}+\varepsilon_{it}$$
where $ISS_{it}$ represents the degree of advanced industrial structure in each province; $X_{control}$ denotes the province characteristic variables, the same as the seven control variables of the benchmark model above. The total effect of technological innovation on provincial carbon emissions is $\frac{\partial CO_{2it}}{\partial G_{it}}$, and the mediating effect of the industrial structure upgrading is $\left(\frac{\partial CO_{2it}}{\partial ISS_{it}}\right)\times \left(\frac{\partial ISS_{it}}{\partial G_{it}}\right)$.

Similarly, the mediation model with energy efficiency as a mediating variable is shown in Equations (4), (7) and (8).
(7)$$EE_{it}=\alpha_{0}+\alpha_{1}\ln G_{it}+\alpha_{2}X_{control}+\delta_{1}W\ln G_{it}+\delta_{2}WX_{control}+\varepsilon_{it}\;\;$$
(8)$$\ln CO_{2it}=\rho_{0}+\rho_{1}\ln G_{it}+\rho_{2}EE_{it}+\sigma_{1}W\ln G_{it}+\sigma_{2}WEE_{it}+\sigma_{3}WX_{control}+\varepsilon_{it}$$
where $EE_{it}$ represents the energy efficiency of each province, and the corresponding mediating effect is $\left(\frac{\partial CO_{2it}}{\partial EE_{it}}\right)\times \left(\frac{\partial EE_{it}}{\partial G_{it}}\right)$.

#### 4.2.1. Variable Selection

##### Dependent Variable

The dependent variable is the amount of Chinese provincial carbon emissions. In order to scientifically characterize the total carbon emissions of each province, eight types of fossil fuels (Eight types of fossil fuels: coal, coke, crude oil, gasoline, kerosene, diesel, natural gas, fuel oil) consumption were selected and converted into 10,000 tons of standard coal, then the IPCC method was introduced for scientific accounting. The calculation is shown in Equation (9). Where *i*, *j*, *t* are provinces, energy categories, and the year in which they are located; *δ* is the coefficient of corresponding energy sources converted into 10,000 tons of standard coal; *Q_ijt_* is the consumption of each energy source; CO_2_ is the total carbon emission (unit: million tons); *β_j_* indicates the CO_2_ emission coefficient corresponding to the eight energy sources. The standard coal and carbon emission coefficients of the eight fossil energy sources are shown in Table 1.
(9)$$CO_{2it}=\overset{8}{\underset{i=1}{\sum }}\delta_{j}\times Q_{ijt}\times \beta_{j}$$

##### Independent Variable

Since we cannot directly observe technological innovation, we proxied technological innovation activities $\left(G_{it}\right)$ as total factor productivity (TFP) [33].

Data envelopment analysis (DEA), a common method for nonparametric approaches, has many advantages. The DEA-Malmquist index method proposed by Färe et al. [34] was used in this study to reflect the relative changes in productivity between different stages to measure technological progress indicators.

Färe defines the distance function as follows:(10)$$D^{t}\left(X^{t},Y^{t}\right)=\inf\left\{\partial >0:\left(X,Y/\partial \right)\in F^{t}\right\}$$
where *X* is the input variable matrix, *Y* is the output variable matrix, ∂ is the output efficiency, and F denotes the frontier technology. Suppose $\left(x^{t},y^{t}\right)$ and $\left(x^{t+1},y^{t+1}\right)$ denote the input and output quantities in periods *t* and *t +* 1, and $D_{i}^{t}\left(\cdot \right)$ and $D_{i}^{t+1}\left(\cdot \right)$ denote the distance functions of different periods with reference to the data in periods *t* and *t +* 1, respectively. For example, $D_{i}^{t}\left(x^{t+1},y^{t+1}\right)$ denotes the distance function for period *t +* 1 when the technical data of period t is used as a reference. According to Caves et al. [35], the Malmquist index for period t and period *t +* 1 can be expressed as:(11)$$M_{i}^{t}=\frac{D_{i}^{t}\left(x^{t},y^{t}\right)}{D_{i}^{t}\left(x^{t+1},y^{t+1}\right)}$$
(12)$$M_{i}^{t+1}=\frac{D_{i}^{t+1}\left(x^{t},y^{t}\right)}{D_{i}^{t+1}\left(x^{t+1},y^{t+1}\right)}$$

Usually, to avoid possible bias in choosing whether period *t* or period *t +* 1 technology is used as a reference, the geometric mean of the $M_{i}^{t}$ and $M_{i}^{t+1}$ indices for periods *t* and *t +* 1 are considered to measure productivity change.
(13)$$M_{i}\left(x^{t+1},y^{t+1};x^{t},y^{t}\right)=\frac{D_{i}^{t}\left(x^{t},y^{t}\right)}{D_{i}^{t+1}\left(x^{t+1},y^{t+1}\right)}\times {\left[\frac{D_{i}^{t+1}\left(x^{t},y^{t}\right)}{D_{i}^{t}\left(x^{t+1},y^{t+1}\right)}\times \frac{D_{i}^{t+1}\left(x^{t+1},y^{t+1}\right)}{D_{i}^{t}\left(x^{t},y^{t}\right)}\right]}^{\frac{1}{2}}$$

Therefore, this paper adopted the DEA-based input-oriented Malmquist index method to measure the total factor productivity of each province in China as a decision unit and converted it uniformly into a cumulative value with 2008 as the base period as the technological innovation indicator. The specific input and output indicators involved in the DEA-Malmquist index measurement are as follows.

(1)Capital Stock of R&D Investment (GI)

Drawing on the research results of Gu and Zhao [36], the capital stock of R&D investment was measured using the perpetual inventory system with the invested indicator. The specific calculation is shown in Equation (14).
(14)$$K_{t}=\left(1-\delta \right)K_{t-1}+RDI_{t}\;K_{0}=RDI_{t}/\left(g+\delta \right)$$
where, *K_t_* and *K_t−_*_1_ denote the R&D capital stock in T and T − 1 periods, respectively, *RDI_t_* is the R&D expenditure in the Tth period, *δ* is the depreciation rate of R&D capital (*δ* was taken as 15%), and *g* is the growth rate of R&D expenditure by the actual growth rate of R&D expenditure corresponding to the sample period. The data are obtained from China Science and Technology Yearbook.

(2)Labor Input (L)

It is generally believed that workers’ quality level and labor time are ideal indicators to measure labor input. However, considering the availability of data, the number of employed populations at the end of the year in each province and city, which is used by most scholars, was selected to represent the amount of labor input. The data are obtained from the statistical yearbooks of each province and city from 2009 to 2020.

(3)Total Output (Y)

Scholars generally use GDP as an indicator of total output, so the GDP of each region from 2008 to 2019 was selected and converted into constant price GDP with 2000 as the base period. The data are obtained from the statistical yearbooks of each province in the calendar year.

#### 4.2.2. Mediating Variables

(1)Industrial Structure Upgrading (ISU)

Scholars generally study industrial structure upgrading from two perspectives: rationalization of industrial structure and advanced industrial structure [37]. In order to scientifically characterize the degree of industrial structure upgrading, this paper selects the advanced industrial structure to measure it. The ratio of the tertiary industry’s value-added to the secondary industry was used to reflect the advanced industrial structure. The larger the ratio is, the more advanced the industrial structure is.

(2)Energy Efficiency (EE)

Some scholars have explored the relationship between technological innovation and energy efficiency [38]. In this study, energy efficiency was measured in terms of energy consumption per unit of output value. The lower the energy input required for the same GDP output, the more efficient the energy utilization, which means that the overall technological progress has a more prominent “green bias”. Conversely, the higher the energy consumed for the same output, the less efficient the energy utilization, which indicates that the overall technological progress is less environmentally friendly.

#### 4.2.3. Control Variables

(1)Urbanization (U)

Urbanization is characterized by the ratio of urban population to total population at the end of the year in the region. The urbanization expansion, infrastructure construction, and housing construction consume many energy-intensive products such as steel and cement, leading to significant carbon emissions. Therefore, urbanization is another cause of environmental degradation.

(2)Foreign Investment (FDI)

There are two hypotheses of foreign investment: “pollution paradise” and “pollution halo”, which are inextricably linked to environmental pollution in host countries. The logarithm of the actual total foreign use in the region was chosen to measure FDI [39], demonstrating the relationship between foreign investment and provincial carbon emissions in China.

(3)Economic Level (PGDP)

The level of economic development has a significant positive impact on carbon emissions [40]. As the circulation of resources and goods accelerates with economic growth, the demand for goods and services by individuals and enterprises expands, which leads to an increase in industrial production activities and ultimately to the rise in carbon emissions. The logarithm of regional GDP per capita was used.

(4)Energy Consumption (EN)

In national economic development, energy consumption mainly relies on fossil fuels, which are also the primary source of carbon emissions. The logarithm of regional total energy consumption was used to characterize energy consumption.

(5)Degree of Marketization (MD)

Technological innovation relies on government intervention in establishing a market-oriented innovation system to optimize resource allocation and promote high-quality economic development. The Fan Gang regional Marketization Index indicated the degree of marketization, which was obtained from the China Marketization Index database.

(6)Population of the Region (POP)

According to the IPAT model (environmental impact = Population × Affluence × Technology) proposed by Ehrlich and Holdren [41], regional human activities can also impact environmental change. Therefore, the regional mid-year population was included in the model as a control variable.

#### 4.2.4. Data Sources

Considering data availability, we used a panel dataset of 30 Chinese provinces (excluding Tibet, Hong Kong, Macao, and Taiwan due to missing data) from 2008 to 2019 to explore the effect of technological innovation on carbon emission reduction. The original data sources are China Statistical Yearbook, China Environmental Statistical Yearbook, China Science and Technology Yearbook, etc. To eliminate possible heteroskedasticity, we used the logarithm terms of carbon emissions (${CO_{2}}_{it}$), foreign investment (FDI), economic level (PGDP), energy consumption (EN), and regional population (POP). Table 2 and Table 3 show the definition, data source, descriptive statistics and pairwise zero-order correlations of the variables. The VIFs are all less than 8, indicating no severe multicollinearity.

## 5. Analysis of the Empirical Results

### 5.1. Spatial Correlation Test

The 0–1 proximity matrix was constructed with geographical proximity to measure the Moran’s I of carbon emissions for 30 Chinese provinces from 2008 to 2019. Table 4 shows the results of the global spatial autocorrelation test of carbon emissions, and all the Moran’s indices of carbon emissions are positive and pass the significance level. It indicates a spatial agglomeration effect of carbon emissions among Chinese provinces.

The Global Moran index only reflects the average correlation but not the spatial correlation of individual provinces and cities. In contrast, Moran’s I scatterplot is used to test spatial correlation. For brevity, only data from 2008 and 2019 were selected as samples. The Local Moran’s I (Moran) scatter plot of carbon emissions shown in Figure 3 was drawn using Stata, where the numbers are the carbon emissions performance in each province or city. The first quadrant (top right) and the third quadrant (bottom left) show the interaction between homogeneous provinces. The first quadrant shows the interaction between high-level provinces or cities and other high-level provinces or cities (i.e., high-high level). The third quadrant shows the interaction between low-level and low-level provinces (i.e., low-low levels). The second quadrant (top left) and the fourth quadrant (bottom right) show interactions between heterogeneous provinces. The second quadrant shows the interaction between low-level and high-level provinces (i.e., low-high level). The fourth quadrant shows the interaction between high-level provinces and low-level provinces or cities (i.e., high-low level). Most provinces are located in the first and third quadrants, and their carbon emissions are both spatially heterogeneous and spatially concentrated, showing a strong spatial agglomeration effect.

### 5.2. Baseline Regression Results

In order to select the suitable spatial econometric model, the residuals were tested for spatial correlation based on the OLS results, shown in Table 5. In general, the SDM model was chosen as the baseline regression model. In addition to Lagrange error, robust error, Lagrange lag, and robust lag are significant. In addition, the Hausman test results showed that the *p*-value was 0, and the original hypothesis was rejected at the 1% significance level. Therefore, the SDM fixed effect was finally selected.

The Wald and likelihood ratio (LR) tests were performed on the SDM. The *p*-values of the Wald and LR spatial lag tests and the spatial error test were 0 at the 1% significance level, indicating that the SDM model had a better fit compared to the other models.

As shown in Table 6, W × CO₂ (rho) in the spatial econometric model is not zero at the 5% level, indicating that carbon emissions have a spatial spillover effect. The coefficient of technological innovation (G) on carbon emissions is −0.083, which passes the 1% significance test, indicating that technological innovation has a significant negative effect on provincial carbon emissions. For each percentage point increase in technological innovation, carbon emission decreases by 0.083%. In addition, energy consumption and urbanization construction are still the driving factors of carbon emissions in control variables. China’s energy consumption structure is still dominated by coal, and the clean energy industry has not yet been able to overturn the market position of fossil fuels. During China’s urbanization, many energy-intensive products such as steel and cement are consumed for infrastructure and housing construction, leading to significant carbon emissions and exacerbating regional carbon emission reduction pressure.

The spatial effect is decomposed using partial differential equations to explore further technological innovation’s spillover effect on provincial carbon emission reduction [42]. As shown in Table 7, the direct effect of technological innovation on provincial carbon emission is significantly negative with a coefficient of −0.095, and the indirect effect coefficient is −0.463, which passes the 1% significance test. Technological innovation promotes not only local carbon emission reduction but also carbon emission reduction in neighboring provinces through the radiation effect. For every 1% increase in technological innovation, the amount of provincial carbon emission is reduced by 0.095%, while the amount of carbon emissions in neighboring provinces is reduced by 0.463%. Therefore, Hypothesis 1 is supported.

To test the robustness of the results, we used the geographic proximity 0–1 spatial weight matrix and alternative measurement of technological innovation (G1) in the spatial econometrics models. We calculated G1 similarly to G but substituted the input-output indicators. The input indicators involve the variables: (1) R&D input capital stock (GI); (2) R&D personnel annual equivalent (RDL); and output indicators involve the variables: (1) patent applications (P); (2) technology market turnover (TY); and (3) total regional output value (Y). As shown in Table 8, the results of both robustness tests are consistent with the main models.

### 5.3. Analysis of the Mediation Mechanism

We examined the mediating role of industrial structure upgrading and energy efficiency improvement to test Hypotheses 2 and 3. Table 9 shows the empirical results of the spatial mediation model.

Models 1–3 in Table 9 were used to test Hypothesis 2. Model 1 contained only the core explanatory variables and control variables. The results of model 1 indicate that technological innovation can effectively contribute to carbon emission reduction. Therefore, Hypothesis 1 is further supported. Model 2 shows that the effect of technological innovation on industrial structure upgrading is not statistically significant. Model 3 shows that industrial structure upgrading and technological innovation both negatively impact carbon emissions, the estimated coefficient of ISU is −0.072.

The spatial intermediation effect of industrial structure upgrading is examined using models 1 to 3. Technological innovation and industrial structure upgrading have a significant effect on carbon emissions ($\beta_{1}$ and $\rho_{2}$ are significant), but technological innovation does not have a significant effect on industrial structure upgrading ($\alpha_{1}$ is not significant), which requires a Sobel test. The Sobel test was applied, and the results did not show a significant mediation effect of industrial structure upgrading between technological innovation and carbon emissions. These results are not entirely consistent with Hypothesis 2. The possible reason is that current Chinese economic sectors are dominated by traditional manufacturing and low-end services. Even though technological innovation leads to a large inflow of labor, resources, and capital into emerging low-carbon and other high-tech industries, it has little impact on reducing the proportion of traditional industries in the economy, thus slowing down the process of technological innovation in promoting industrial structure transformation. Yet, the coefficient of industrial structure upgrading is negative and significant, which partially supports our Hypothesis 2 that industrial structure upgrading promotes carbon emission reduction.

Models 1, 4, and 5 in Table 9 were used to test Hypothesis 3. Based on the significant contribution of technological innovation to carbon emission reduction in model 1, the effect of energy efficiency is further considered in model 5. The results show that energy efficiency is positively related to carbon emissions, where the less energy used per GDP unit, the less amount of carbon emissions it may cause. Model 4 analyzes the relationship between technological innovation and energy efficiency. The results show that technological innovation is significantly and negatively correlated with energy efficiency, indicating that technological innovation substantially improves energy efficiency.

The spatial intermediation effect model of energy efficiency was constructed using models 1, 4, and 5. The coefficients of the effects of technological innovation and energy efficiency in the model are significant, indicating that energy efficiency plays a mediating role in technological innovation affecting carbon emissions. Technological innovation is negatively related to carbon emission via energy use efficiency. The improvement of technology can improve energy utilization efficiency and ultimately promote carbon emission reduction. These results verify Hypothesis 3.

Figure 4 shows the estimated coefficients of the theoretical model. Technological innovation can significantly reduce carbon emissions, and for every 1% increase in technological innovation, the amount of carbon emissions will be reduced by 0.083%. Both industrial structure and energy efficiency play a mediating role. Among them, technological innovation can significantly reduce carbon emissions by improving energy efficiency, while the path of carbon reduction by upgrading industrial structure had not yet been formed.

### 5.4. Heterogeneity Analysis

Economic development in China’s eastern, central, and western regions is difficult to analyze due to differences in geographic location and early policy preferences [29]. Therefore, this research classifies 30 Chinese provinces according to the significant differences in regional economic development and geographic location in eastern, central, and western regions and provides an in-depth analysis of the relationship between technological innovation and regional differences in carbon emissions. Among them, Guangdong, Fujian, Zhejiang, Jiangsu, Shandong, Shanghai, Beijing, Tianjin, Hebei, Liaoning, and Hainan are located in the eastern region; the central region includes Hubei, Hunan, Henan, Anhui, Jiangxi, Shanxi, Jilin, and Heilongjiang; the western region includes Xinjiang, Qinghai, Gansu, Ningxia, Yunnan, Guizhou, Sichuan, Shaanxi, Chongqing, Guangxi, and Inner Mongolia.

As shown in Table 10, the effect of technological innovation on carbon emissions shows significant spatial heterogeneity. The direct impact of technological innovation on carbon emissions and the indirect effect via industrial structure upgrading is only significant in the western regions (Figure 5c). In the eastern region, the indirect effect of technological innovation on carbon emissions via energy efficiency improvements is significant (Figure 5a). In the central region, technological innovation has little impact on carbon emission, neither directly nor indirectly via industrial structure upgrades nor energy efficiency improvements (Figure 5b).

One of the possible reasons that the effects show significant regional heterogeneity is that the eastern, central, and western regions are subject to substantial historical legacy and economic development modes. The eastern region has a more open economic and political environment, and resources such as capital, technical capabilities, and talents are more advanced among the three regions. Meanwhile, heavy industries in the eastern regions have been relocated to the other areas, and the primary economic focus of the region is high-value-added and high-tech industries. Thus, the industrial structure in the eastern region is more rational and advanced than in the other two regions in China [37]. However, due to its enormous economic size and economic activities, energy consumption in the eastern region is considerable, leading to more carbon emissions. Therefore, technological innovation in energy efficiency is the primary path to controlling carbon emissions and maintaining economic growth in the eastern region.

The central region is the agricultural base, the energy and raw material base, and the modern equipment manufacturing and high-tech industrial base. The central region has an aggregation of advantageous industries and a complete industrial chain, but the industrial structure has been “heavy” for a long time. Thus, significant and continuous investment in technological innovation is required to upgrade industrial structures and reduce carbon emissions. However, unbalanced and insufficient development is still a major challenge.

Due to the low concentration of pillar industries and low inter-industry coordination ability, the industrial cluster of the western region is still in its infancy and is generally scattered in various locations. The carbon emissions in the western region are also lower than that of the other two regions due to fewer industrial activities and unique geographical conditions. In terms of economic development, industrial clusters are not only conducive to improving the overall competitiveness of the western region but also help to strengthen the effective collaboration between firms in the cluster and could remarkably promote the adjustment and upgrading of the industrial structure in the western region. Therefore, technological innovation in industrial development and structure upgrading could be the primary path to the western region’s carbon control and economic development.

In summary, technological innovation affects regional carbon emissions through various mechanisms, supporting Hypothesis 4.

## 6. Conclusions

Technological innovation plays an essential role in achieving China’s “double carbon” target and deserves more attention. Understanding the mechanism and effect of technological innovation in provincial carbon reduction will be helpful in the development of carbon reduction policies and approaches. In this study, we studied 30 Chinese provinces, considering the direct and spatial spillover effects. We also explored the intermediary role of industrial structure upgrading and energy efficiency improvement.

The main conclusions and contributions are summarized as follows. First, technological innovation not only helps with carbon emission reduction in a province but also helps to promote carbon emission reduction in surrounding areas. Second, technological innovation could help reduce carbon emissions via improving energy efficiency in energy consumption, while the mediating effect of industrial structure upgrading is not yet significant. Finally, the impact of technological innovation on carbon emission reduction is of significant regional heterogeneity. In the eastern region, technological innovation promotes carbon emission reduction through energy efficiency improvement. Still, the effect of technological innovation on carbon emission reduction via industrial structure upgrades is not significant. In the central region, regional carbon emission reduction is under pressure from the industrial transfer. Sufficient and continuous investment in technological innovation may provide opportunities for carbon emission reduction through industrial changes and clean energy substitution. In the western region, technological innovation can directly promote carbon emission reduction and carbon emission reduction through industrial structure upgrading.

## 7. Research Implications

### 7.1. Implications

The following policy implications are proposed to help decision-makers in China and in other developing countries adjust to more appropriate carbon control and technological policies. First, local governments should notice the spatial spillover effect of technological innovation on reducing provincial carbon emissions and establish knowledge exchange and communication mechanisms to enhance technical collaboration in green and low-carbon technologies. It is necessary to strengthen communication and cooperation among neighboring provinces through regular meetings and communication to ensure the synergy effect of technology and carbon control policies. Provinces with a high carbon emission level should play a leading role in setting up regular information exchange with surrounding regions on advanced technologies.

Second, technological investments in upgrading regional industrial structure and improving energy use efficiency will both strengthen the effect on carbon emission reduction. Improving the technology innovation system and promoting the innovation-driven strategy in economic development and industrial activities will lead to a greener development mode. The incentive mechanism for technological innovation in regional industrial structure and energy efficiency and providing subsidies for adopting clean energy could help industrial sectors transform toward low-emission structures and increase energy efficiency—building R&D investment platforms and guiding investors on the direction of technology trends to help develop green technologies. Adhering to market orientation and optimizing the flow of innovation factors, policies should highlight the leading position of industrial firms in innovative activities and encourage the commercialization of low-carbon technologies.

Finally, policymakers should develop unique carbon control and environmental regulations according to local conditions due to regional differences in the effects and mechanism of technological innovation on carbon emission reduction. Since historical legacy formed the current regional industrial and energy structures, local decision-makers should adjust their regulations and carbon governance accordingly. For instance, governments in the eastern region should optimize factor allocation and support low-carbon and green energy consumption along with promoting the commercialization of energy-saving technology. The central region should gradually improve the industrial structure, promote the development of low-end manufacturing industries to the middle and high-end, and fully use information technology such as big data and the Internet of Things to build technical capabilities. The western region should leverage its comparative advantage of abundant resources to promote the coordinated development of advanced technologies and industries, thus supporting the green and sustainable development of the regional economy and society.

### 7.2. Limitation

This paper focuses on the impact of technological innovation on provincial carbon emissions in China. The spatial mediated effect model sheds light on the mechanisms through which technological innovation promotes carbon emission reduction. The heterogeneity analysis of different regions provides guidance for developing local carbon control policies. However, this study has a few limitations that need to be further considered. First, this study does not consider non-desired output when calculating total factor productivity as an indicator of technological innovation. Second, we used a province-level panel dataset because of city-level data unavailability. Finally, we did not consider the impact of current environmental regulations on carbon emission. Environmental regulations are critical to regional industrial and economic development and need further investigation along with carbon emission reduction.

## Figures and Tables

**Figure 1 ijerph-19-09543-f001:**
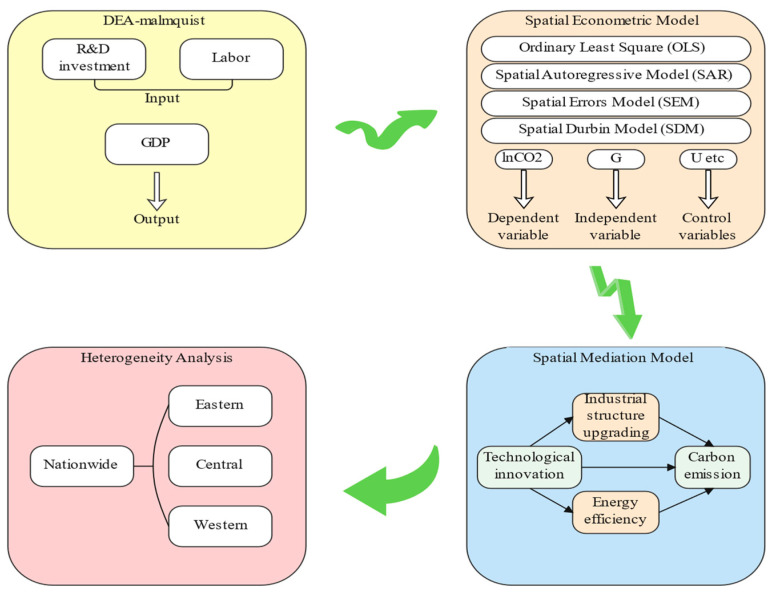
Analysis framework.

**Figure 2 ijerph-19-09543-f002:**
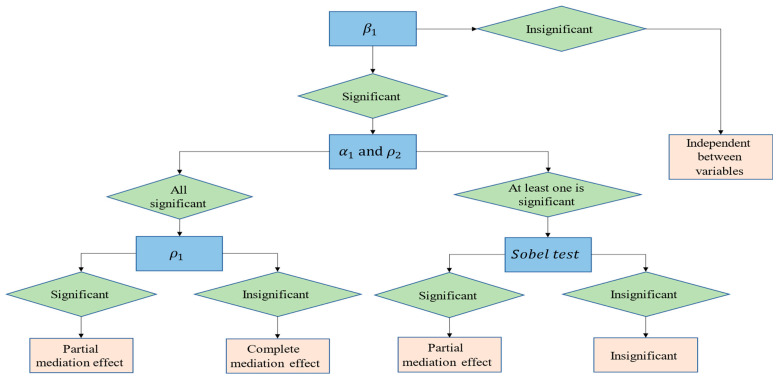
Flowchart of mediating effect test.

**Figure 3 ijerph-19-09543-f003:**
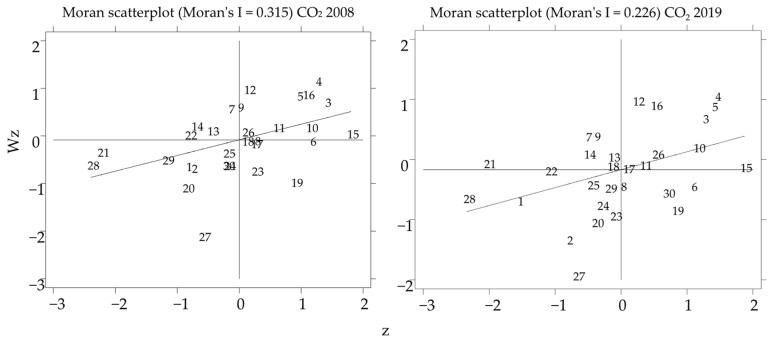
Moran’s I scatter plot of carbon emission.

**Figure 4 ijerph-19-09543-f004:**
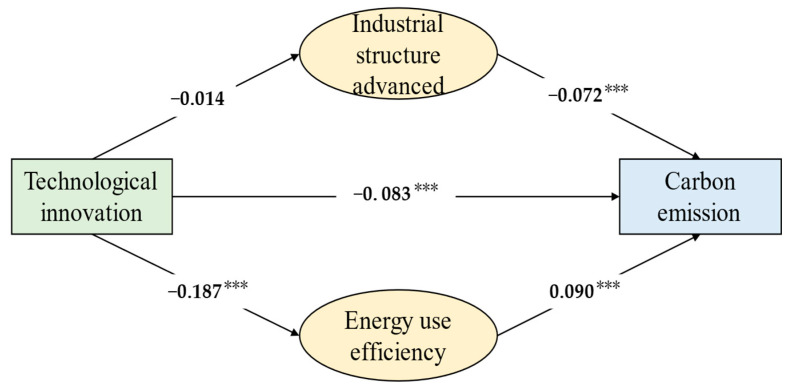
The impact of technology innovation on carbon emissions. Note: *: *p* < 0.1, **: *p* < 0.05, ***: *p* < 0.01.

**Figure 5 ijerph-19-09543-f005:**
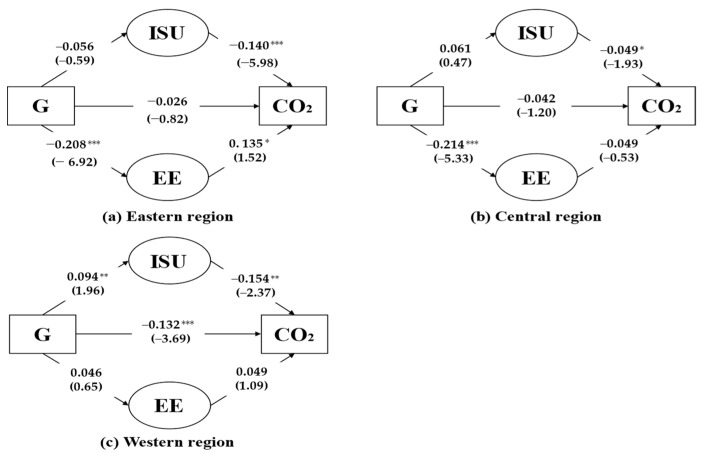
The effect of technological innovation on carbon emissions in eastern, central, and western regions. Note: *: *p* < 0.1, **: *p* < 0.05, ***: *p* < 0.01. z-values in parentheses.

**Table 1 ijerph-19-09543-t001:** Eight types of fossil energy standard coal and carbon emission factors.

Types	Unit	Standard Coal	Carbon Emission Factor
Raw coal	kg of standard coal/kg	0.7143	0.7476
Coke	kg of standard coal/kg	0.9700	0.1128
Crude oil	kg of standard coal/kg	1.4300	0.5854
Gasoline	kg of standard coal/kg	1.4700	0.5532
Kerosene	kg of standard coal/kg	1.4717	0.3416
Diesel fuel	kg of standard coal/kg	1.4600	0.5913
Natural gas	t standard coal/million cubic meters	13.3000	0.4479
Fuel oil	kg of standard coal/kg	1.4286	0.6176

**Table 2 ijerph-19-09543-t002:** Definition and data source of the variables.

Variable		Definition	Data Source
Dependent variable	CO_2_	The amount of Chinese provincial carbon emissions	China Environmental Statistical Yearbook
Independent variables	G	Total factor productivity	China Science and Technology Yearbook
Mediating variables	ISU	The ratio of the tertiary industry’s value-added to the secondary industry	China Statistical Yearbook
	EE	The terms of energy consumption per unit of output value	China Environmental Statistical Yearbook
Control variables	U	The ratio of urban population to total population at the end of the year in the region	China Statistical Yearbook
	FDI	The actual total foreign use in the region	China Statistical Yearbook
	PGDP	Regional GDP per capita	China Statistical Yearbook
	EN	Regional total energy consumption	China Statistical Yearbook
	MD	Regional Marketability Index	China Marketization Index database
	POP	The regional mid-year population	China Statistical Yearbook

**Table 3 ijerph-19-09543-t003:** Descriptive statistics and pairwise zero-order correlations.

Variables	Obs	Mean	S.D.	Min	Max	(1)	(2)
(1) lnCO₂	360	9.086	0.744	7.036	10.655		
(2) G	360	1.121	0.313	0.463	2.416	0.093 *	
(3) ISU	360	1.101	0.635	0.499	5.169	−0.315 ***	0.026
(4) EE	360	1.431	0.734	0.401	3.928	−0.049	−0.238 ***
(5) lnPGDP	360	10.141	0.476	8.739	11.354	−0.252 ***	−0.042
(6) lnFDI	360	12.719	1.649	7.310	15.086	0.473 ***	0.219 ***
(7) U	360	55.521	13.186	29.112	89.632	−0.019	0.237 ***
(8) lnEN	360	9.377	0.672	7.034	10.625	0.926 ***	0.188 ***
(9) MD	360	6.445	1.926	2.33	11.4	0.288 ***	0.318 ***
(10) LnPOP	360	8.181	0.768	4.117	9.352	0.692 ***	0.126 **
Variables	(3)	(4)	(5)	(6)	(7)	(8)	(9)
(1) lnCO₂							
(2) G							
(3) ISU							
(4) EE	−0.304 ***						
(5) lnPGDP	−0.055	0.108 **					
(6) lnFDI	0.119 **	−0.743 ***	−0.259 ***				
(7) U	0.567 ***	−0.439 ***	−0.261 ***	0.494 ***			
(8) lnEN	−0.242 ***	−0.203 ***	−0.237 ***	0.571 ***	0.068		
(9) MD	0.233 ***	−0.755 ***	−0.184 ***	0.809 ***	0.677 ***	0.439 ***	
(10) LnPOP	−0.227 ***	−0.462 ***	−0.093 *	0.595 ***	−0.141 ***	0.80 ***	0.419 ***

Note: *: *p* < 0.1, **: *p* < 0.05, ***: *p* < 0.01.

**Table 4 ijerph-19-09543-t004:** Global Moran’s index CO_2_ under spatial distance weight matrix.

Year	I	Year	I	Year	I
2008	315 ***	2012	280 **	2016	252 **
2009	295 ***	2013	274 **	2017	241 **
2010	296 ***	2014	263 **	2018	243 **
2011	297 **	2015	266 **	2019	226 **

Note: *: *p* < 0.1, **: *p* < 0.05, ***: *p* < 0.01.

**Table 5 ijerph-19-09543-t005:** Model selection.

Indicator	Statistic	*p*-Value	Indicator	Statistic	*p*-Value
LM-Spatial_Lag	29.052	0.000	Wald-Spatial_Lag	92.16	0.000
Robust-LM-Spatial_Lag	38.240	0.000	LR-Spatial_Lag	81.71	0.000
LM-Spatial_Erro	0.002	0.969	Wald-Spatial_Erro	75.53	0.000
Robust-LM-Spatial Erro	9.190	0.002	LR-Spatial_Erro	76.57	0.000

**Table 6 ijerph-19-09543-t006:** Benchmark regression results.

CO₂	Coef.	Std. Err.	Z	P > z	Confidence Interval	
Main						
G	−0.083	0.021	−3.960	0.000	−0.123	−0.042
lnPGDP	0.005	0.009	0.530	0.594	−0.012	0.022
U	0.006	0.002	2.590	0.010	0.001	0.009
lnFDI	−0.005	0.009	−0.580	0.563	−0.022	0.012
lnEN	1.159	0.059	19.540	0.000	1.043	1.275
MD	0.025	0.009	2.900	0.004	0.008	0.042
lnPOP	0.004	0.015	0.31	0.758	−0.024	0.033
W×						
G	−0.374	0.056	−6.620	0.000	−0.484	−0.263
lnPGDP	0.019	0.018	1.120	0.261	−0.015	0.054
U	−0.006	0.004	−1.290	0.197	−0.014	0.003
lnFDI	0.069	0.013	5.190	0.000	0.043	0.096
lnEN	−0.202	0.138	−1.460	0.143	−0.473	0.068
MD	−0.040	0.016	−2.540	0.011	−0.071	−0.009
lnPOP	0.012	0.030	0.410	0.682	−0.047	0.072
Spatial						
rho	0.177	0.084	2.100	0.036	0.012	0.341
Sigma2_e	0.004	0.001	13.38	0.000	0.003	0.004

**Table 7 ijerph-19-09543-t007:** Direct effect, indirect effect, and the total effect of SDM.

Variables	Direct Effect		Indirect Effect		Total Effect	
	Coef.	Std. Err.	Coef.	Std. Err.	Coef.	Std. Err.
G	−0.095 ***	0.022	−0.463 ***	0.078	−0.558 ***	0.084
lnPGDP	0.005	0.009	0.024	0.020	0.029	0.023
U	0.005 ***	0.002	−0.005	0.005	0.001	0.004
lnFDI	−0.002	0.008	0.082 ***	0.016	0.080 ***	0.017
lnEN	1.161 ***	0.056	−0.007	0.110	1.154 ***	0.095
MD	0.024 ***	0.008	−0.044 **	0.018	−0.020	0.018
lnPOP	0.005	0.015	0.017	0.037	0.022	0.044

Note: *: *p* < 0.1, **: *p* < 0.05, ***: *p* < 0.01.

**Table 8 ijerph-19-09543-t008:** Results of robustness tests.

Effect	Weight Substitution		Measurement Substitution	
	Coef.	Std. Err.	Coef.	Std. Err.
Direct effect	−0.067 ***	0.022	−0.031 *	0.018
Indirect effect	−0.402 ***	0.072	−0.211 ***	0.064
Total effect	−0.469 ***	0.078	−0.243 ***	0.071

Note: *: *p* < 0.1, **: *p* < 0.05, ***: *p* < 0.01.

**Table 9 ijerph-19-09543-t009:** Results of the spatial mediation model.

Variables	Model 1	Model 2	Model 3	Model 4	Model 5
	lnCO_2_	ISU	lnCO_2_	EE	lnCO_2_
G	−0.083 ***	−0.014	−0.086 ***	−0.187 ***	−0.064 ***
ISU			−0.072 ***		
EE					0.090 ***
lnPGDP	0.005	0.006	0.004	0.024	0.003
U	0.006 ***	−0.031 ***	0.003	−0.018 ***	0.007 ***
lnFDI	−0.005	0.039 *	−0.002	−0.035 **	−0.002
lnEN	1.159 ***	−0.345 **	1.132 ***	1.059 ***	1.068 ***
MD	0.025 ***	−0.038 *	0.023 ***	−0.047 ***	0.028 ***
lnPOP	0.004	−0.079 **	−0.001	−0.017	0.005
Fixed effect	YES	YES	YES	YES	YES
obs	360	360	360	360	360
R^2^	0.8307	0.6712	0.8366	0.7187	0.8355

Note: *: *p* < 0.1, **: *p* < 0.05, ***: *p* < 0.01.

**Table 10 ijerph-19-09543-t010:** Heterogeneity results of the eastern, central, and western regions.

Regions	Direct Effect	Indirect Effect	Total Effect
Eastern Region	−0.029	−0.154 **	−0.184 **
Central Region	−0.027	−0.092 ***	−0.118 **
Western Region	−0.141 ***	−0.146	−0.287

Note: *: *p* < 0.1, **: *p* < 0.05, ***: *p* < 0.01.

## Data Availability

Data and materials are available from the authors upon request.

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
