# Peer review of "Carbon Dioxide Emissions Reduction through Technological Innovation: Empirical Evidence from Chinese Provinces"

_ijerph, 2022, doi:10.3390/ijerph19159543_

Round 1
Reviewer 1 Report
This is a good data set to use as it covers a good long time period as well as it appears 30 of China’s 31 provinces with meaningful results.
References are good, relevant and as far as I am aware some of the most recent.
Hypothesis 1 should be written as two separate hypothesese can be written as 1a and 1b i.e. separately tested then can do as one but separate initially.
Under H2 the first sentence is a bit confusing and seems to be contradictory.
H3 seems to be saying that reductions in emissions due to technological innovation is reduced by energy efficiency – is this what it is meant to be saying?
Section 4.1 please explain each of the variables in the equations i.e. a table perhaps here or at the end of the paper and referred to here.
Perhaps more clarification on the difference between equations 1 and 3 in section 4.1
Figure 2 is good but where is the discussion on it?
Do you have references for the use of total factor productivity for technological innovation activities (p.8)?
Control variables are per province? Make sure it is clear i.e. while the definitions are good just clarify this part e.g. in foreign investment but check all that it is clear.
Can you clarify Figure 4 please? It is good but needs better discussion.
An interesting paper with interesting results.
Reviewer 2 Report
The authors seek to determine to what extent technological innovation influences carbon dioxide emissions. This is a very sophisticated analysis both conceptually and empirically.
At the conceptual level, the authors consider numerous effects. Among these are geographic spillover effects, and how the relationship between emissions and innovations is mediated through a variety of factors such as energy efficiency and industrial structure upgrades. As a result, the authors emerge from the introductory portion of the manuscript with four testable hypotheses which they state clearly.
The authors then spend a good portion of their manuscript revealing various regression equations they are considering, along with the rationale for each. Their awareness of problems common in regression such as heteroskedasticity and autocorrelation is a refreshing change from many papers I have seen. They give descriptions of techniques and specifications designed to prevent these problems from disturbing the results of their analysis.
Once they have completed that discussion, they go on to discuss variables and data sources. These discussions are extensive. The authors address all the relevant issues.
Although section 5 is labelled results, the authors continue to present results in sections 6 and 7 as well (on mediation mechanisms and heterogeneity analysis). I really think this numbering should be changed. Sections 6 and 7 should be a part of results (Sections 5.3 and 5.4 respectively). Then the Conclusions should be section 6, not section 8.
The findings and the conclusions are sound. I am not asking the authors to change any content. But I am asking for a minor revision. The manuscript should contain 6 sections, not 8. If you are going to label a section “results” you should not be including actual results on hypotheses in later sections. The results should be in the results section. Use subsections instead. This is only a minor revision.
